# Role of the Hedgehog Pathway and CAXII in Controlling Melanoma Cell Migration and Invasion in Hypoxia

**DOI:** 10.3390/cancers14194776

**Published:** 2022-09-29

**Authors:** Gaia Giuntini, Federica Coppola, Alessandro Falsini, Irene Filippi, Sara Monaci, Antonella Naldini, Fabio Carraro

**Affiliations:** 1Department of Molecular and Developmental Medicine, Cellular and Molecular Physiology Unit, University of Siena, 53100 Siena, Italy; 2Harvard School of Dental Medicine, Oral Medicine, Infection and Immunity, Boston, MA 02115, USA; 3Department of Medical Biotechnologies, Cellular and Molecular Physiology Unit, University of Siena, 53100 Siena, Italy

**Keywords:** hypoxia, hedgehog pathway, carbonic anhydrase, motility, metalloproteinases, melanoma, cancer

## Abstract

**Simple Summary:**

Malignant melanoma is the leading cause of death among skin cancer patients due to its tendency to metastasize. Hypoxia, which is a common feature of the tumor microenvironment, as well as different alterations at the molecular level, may affect melanoma aggressiveness. The aims and the objectives of this work were to investigate whether and how the Hedgehog pathway and CAXII may control malignant melanoma cell migration and invasiveness either in normoxic or hypoxic conditions. To this end we evaluated the migratory and invasive capabilities of SK-MEL-28 and A375 cell lines, where the hedgehog pathways and CAXII were inhibited by short interfering RNA. Our results indicate that SMO and GLI1 silencing caused the downregulation of CAXII expression. Furthermore, the Hedgehog pathway and CAXII inhibition, resulted in impaired melanoma cell migration and invasion either under normoxic or hypoxic conditions. The fact that CAXII and the Hedgehog pathway are relevant in melanoma cell invasion may be exploited to discover novel and promising therapeutical targets for melanoma clinical management.

**Abstract:**

Background: Malignant melanoma is the leading cause of death among skin cancer patients due to its tendency to metastasize. Alterations at the molecular level are often evident, which is why melanoma biology has garnered increasing interest. The hedgehog (Hh) pathway, which is essential for embryonic development, is aberrantly re-activated in melanoma and may represent a promising therapeutic target. In addition, carbonic anhydrase XII (CAXII) represents a poor prognostic target for hypoxic tumors, such as melanoma, and is involved in cell migration. Thus, we decided to investigate whether and how the Hh pathway and CAXII may control melanoma cell migration and invasiveness. Methods: The migratory and invasive capabilities of SK-MEL-28 and A375 cell lines, either un-transfected or transiently transfected with Smoothened (SMO), GLI1, or CAXII siRNA, were studied under normoxic or hypoxic conditions. Results: For the first time, we showed that SMO and GLI1 silencing resulted in the downregulation of CAXII expression in both moderately and highly invasive melanoma cells under hypoxia. The Hh pathway as well as CAXII inhibition by siRNA resulted in impaired malignant melanoma migration and invasion. Conclusion: Our results suggest that CAXII and the Hh pathway are relevant in melanoma invasion and may be novel and promising therapeutical targets for melanoma clinical management.

## 1. Introduction

Melanoma, a heterogeneous disease arising from the malignant transformation of melanocytes, is rare, but it is the deadliest form of skin cancer. Ultraviolet radiation exposure is considered to be the major etiological agent in the development of melanoma, but hereditary or genetic predisposition can also be responsible [1]. Alterations at the molecular level are often evident, which is why melanoma biology has garnered increasing interest in the last two decades. Indeed, many driver mutations related to intracellular pathways controlling proliferation (BRAF, NRAS, NF1); resistance to apoptosis (TP53); and cell cycle inhibition (CDKN2A) have been already identified [2], and, based on this evidence, several chemical inhibitors, especially BRAF inhibitors, have been investigated in clinical trials. However, studies on their long-term clinical benefits have reported a wide range of drug resistance [3,4,5,6]. In this scenario, it is necessary to further investigate the molecular mechanisms that drive malignant melanoma genesis, progression, and metastasis, in order to employ them as new possible therapeutic targets. In this regard, the hedgehog pathway, an essential embryonic developmental pathway, seems to be a promising candidate. 

The seven-transmembrane domain protein Smoothened (SMO) is the main signal transducer of the canonical Hh pathway. The effectors of the Hh pathway in mammals are the glioma-associated oncogenes (GLI1–3). GLI1 is an activator, and it is only present in its full-length form, when the pathway is active, while GLI2 and GLI3 can be either activators or repressors based on their proteolytic processing.

On the other hand, the noncanonical SMO-independent activation of GLI can occur in the absence of Hh ligand binding to the 12-pass transmembrane protein receptor Patched (PTCH), as GLI activation is regulated by several other oncogenic pathways and signaling proteins external to the Hh pathway; this route of GLI activation is exclusively ligand-independent. Accumulating evidence has implicated both routes of GLI regulation in the development of many known cancers, indicating the role of Hh in cancer progression [7]. Furthermore, evidence exists regarding its involvement in melanoma, and researchers agree on the fact that aberrant Hh reactivation is responsible for melanoma cancer stem cell (CSC) proliferation, survival, and self-renewal [8,9,10,11]. Indeed, new Hh inhibitors have been designed in recent years to impair these melanoma features [12,13,14,15,16,17], with the aim of discovering new therapeutic strategies, but further studies are required to investigate the involvement of Hh in melanoma progression and metastasis. Melanoma severity is not represented by the primary tumor, for which the gold-standard management is local resection, but by its tendency to migrate, enter the blood or lymphatic circulation, and metastasize [18]. Unfortunately, even nowadays, metastatic melanoma makes treating patients difficult, and prognosis is often uneven [18,19]. For this reason, blocking metastasis is the mainstay for melanoma researchers, whose aim is to design a single-patient precision therapy targeting the molecular pathways involved in melanoma dissemination [20,21,22,23,24,25]. In addition to molecular heterogeneity, melanoma also presents a great complexity of both primary and metastasis tumor microenvironments (TMEs), which have been shown to comprise a variety of cell types; extracellular matrix (ECM) elements; and specific biophysical properties, such as hypoxia and acidification [26,27]. Hypoxia is defined as a low-oxygen (O_2_) tension state characterized by the induction of hypoxia-inducible factors (HIF). Hypoxia often leads normal cells to apoptosis. Melanoma cells can adapt to the adverse microenvironment of low-O_2_ tension, proliferate, and turn into an even more malignant phenotype [28,29,30]. In fact, melanoma metastasis often originates from the low-O_2_ levels of the primary tumor [31], which is in accordance with the evidence that the presence of hypoxia within the primary tumor mass is associated with poor prognosis in patients [30]. Thereafter, cancer cells tend to reprogram their metabolic activities to increase glycolysis under normal levels of O_2_ (called “aerobic glycolysis”), and this contributes to transforming the final product of the glycolytic cascade, pyruvate, into lactate, which is exported to the extracellular space and acidifies the TME [32]. This phenomenon, also called the Warburg effect, is enhanced under hypoxic conditions and contributes to the exacerbation of the extracellular pH (pHe). In this context, it has been documented that carbonic anhydrases (CAs) play a pivotal role in regulating pH homeostasis through their ability to catalyze the reversible hydration of carbon dioxide to bicarbonate [33]. Five different families encode these metalloproteins, and 15 isoforms in human CAs have been described [34,35]. However, the isoforms CAIX and CAXII have been the subject of several studies in the last few years, as they are associated with tumor hypoxia and often poor prognosis [36,37,38,39,40,41,42,43]. In fact, in previous papers, we suggested a link between the Hh pathway and CAXII expression in breast cancer and melanoma cell lines, since their pharmacological inhibition resulted in an impairment of cell migration and invasion [44,45]. Considering the morbidity and invasiveness of melanoma, there is still an ongoing interest in the identification of the mechanism underlaying the role of CAXII in melanoma cells. Therefore, in this study, we used an siRNA approach to target the upstream SMO and the downstream GLI1 signaling of the Hh pathway to evaluate the potential modification of CAXII expression and its correlation with the migration and invasiveness of two melanoma cells lines, which differed in terms of their invasive capabilities. We demonstrated that the inhibition of both SMO and GLI1 decreased CAXII expression and impaired melanoma cell migration and invasion.

## 2. Results

### 2.1. Effects of Hypoxia on Untreated SK-MEL-28 and A375

During tumor development and progression, cancer cells often have restricted access to nutrients and oxygen [1]. Since the hypoxic response is mainly ascribed to HIFs, we firstly verified HIF-1α expression after 24 h exposure under normoxia or hypoxia. As shown in Figure 1A, HIF-1α protein expression was significantly increased in both cell lines at 2% O_2_, a pO_2_ which resembled the hypoxic tumor microenvironment (Figure 1A). At the same time, we determined the protein expression of SMO, GLI1, and CAXII in both cell lines. Hypoxia significantly reduced the expression of GLI1 and SMO in the SK-MEL-28 cell line, while only GLI1 was significantly reduced in A375 (Figure 1B). In contrast, we observed an increased CAXII expression, which was highly significant in A375. Furthermore, we investigated whether hypoxia could affect the migration and invasion of both cell lines. As shown in Figure 1C, neither 24 or 48 h of hypoxic exposure significantly affected the migratory capability of both cell lines. In contrast, and in accordance with the CAXII results, the A375 cell line invasiveness was significantly increased by hypoxic treatment, while SK-MEL-28 cells were not affected (Figure 1D).

### 2.2. SMO, GLI1, and CAXII Transient Knockdown in SK-MEL-28

To evaluate the potential impact of both the SMO-dependent and SMO-independent regulation of GLI1 and CAXII on the migratory and invasiveness capabilities, we performed experiments using siRNA for their transiently knockdown. First of all, we determined the efficiency of the siRNA approach in the SK-MEL-28 cell line, under both normoxia and hypoxia, by qRT-PCR and western blotting. As shown in Figure 2A,B, SMO was significantly inhibited under both normoxia and hypoxia at both mRNA and protein levels. A similar inhibition was observed for the GLI1 siRNA experiments, as shown in Figure 2C,D. Notably, CAXII inhibition was observed only under hypoxia (Figure 2E,F). This may be explained by the fact that, as expected, under normoxic conditions CAXII expression was almost undetectable, as shown above in Figure 1B. 

### 2.3. SMO, GLI1, and CAXII Transient Knockdown in A375

Similarly to the SK-MEL-28 experiments, we transiently knocked down SMO, GLI1, and CAXII in the A375 cell line. As shown in Figure 3A–C, SMO expression was significantly reduced by siRNA under both normoxic and hypoxic conditions. A similar inhibition was observed for the GLI1 siRNA experiments (Figure 3D–F). However, we could not observe a consistent reduction in both SMO and GLI1 protein levels under hypoxia, which could be explained by the fact that hypoxia itself resulted in an SMO and GLI1 decrease (see Figure 1A). Regarding CAXII siRNA, we observed a significant inhibition under both mRNA and protein levels and under both normoxia and hypoxia, indicating the achievement of an efficient transient knockdown (Figure 3G–I). More interestingly, the CAXII siRNA treatment resulted in a significant downregulation in both GLI1 and SMO for both mRNA and protein levels under hypoxia (Figure 3 J–L). The latter results indicate that when CAXII was inhibited, Hh activation may have been reduced, suggesting potential crosstalk between CAXII and Hh in hypoxic melanoma cells.

### 2.4. SMO and GLI1 Transient Knockdown Reduced CAXII Protein Levels

Previous results have suggested crosstalk between Hh and CAXII in breast cancer cell lines [44,46]. To test whether a similar interaction is present in melanoma cells, we next determined whether CAXII expression was affected by SMO or GLI1 in both the SK-MEL-28 and A375 cell lines. As shown in Figure 4A,B, SMO transient knockdown significantly inhibited the protein expression of CAXII in both cell lines. Notably, GLI1 siRNA treatment resulted in a significant downregulation in CAXII only under hypoxia (Figure 4C,D). This was in accordance with the results regarding CAXII expression in A375 cells, which was significantly higher only under hypoxic conditions (see Figure 1B above). 

### 2.5. SK-MEL-28 Cell Migration Is Impaired by CAXII, SMO, and GLI1 Transient Knockdown

As the transient knockdown of CAXII, SMO, and GLI1 was successful, we next investigated its functional impact on SK-MEL-28 cell migration. To this end, we performed a wound-healing assay and monitored the cells’ ability to repair wounds up to 48 h, under both normoxia and hypoxia. As shown in Figure 5A, CAXII inhibition impaired SK-MEL-28 migration at both 24 and 48 h. This observation was significant under both normoxia and hypoxia. A similar trend was obtained for the inhibition of the Hh pathway (Figure 5B). These results indicated that Hh is involved in melanoma cell migration and confirmed its crosstalk with CAXII in this cell type. 

### 2.6. A375 Cell Migration Is Inhibited by CAXII, SMO, and GLI1 siRNA

To better understand the impact of CAXII and the Hh pathway CAXII in melanoma cell migration, we performed a scratch assay in the A375 cell line.

As shown in Figure 6A, CAXII inhibition affected A375 cell migration under both normoxia and hypoxia. However, the decrease was significant only at 48 h under both conditions. On the other hand, and similarly to the SK-ML-28 cell line, Hh pathway inhibition impaired A375 cell migration at both 24 and 48 h under both normoxia and hypoxia (Figure 6B). These results suggested once again the involvement of the Hh pathway, along with CAXII, in melanoma cell migration. 

### 2.7. SMO and GLI1 Transient Knockdown Impaired Melanoma Cell Invasion

Based on the cell migration results, we decided to investigate whether the Hh pathway was involved in melanoma cell invasion. To this end, we performed a modified Boyden chamber assay using both the SK-MEL-28 and A375 cell lines, wherein SMO and GLI1 were transiently knocked down by siRNA. As shown in Figure 7A, SK-MEL-28 invasion was significantly reduced by both SMO and GLI1 siRNAs after 24 h under normoxia and hypoxia. Similar results were obtained for the A375 cell line, though the invasion capability under hypoxia significantly increased (Figure 7B). Indeed, although this cell line was characterized by a lower cell migration capability, it presented a higher level of invasive behavior when compared to SK-MEL-28 [46]. 

### 2.8. CAXII Transient Knockdown Resulted in Decreased Cell Invasion and MMP-2/ MMP-9 Activity

As CAXII inhibition resulted in a decreased migratory capability in both melanoma cell lines, we next determined whether CAXII-targeted siRNA resulted in an inhibition of SK-MEL-28 and A375 cell invasion under either normoxia or hypoxia. As shown in Figure 8A, the SK-MEL-28 invasion capability was significantly reduced under both conditions. Similar results were obtained for the A375 cell lines (Figure 8B). Notably, and in contrast to the SK-MEL-28 cells, where hypoxia did not affect cell invasion, the A375 invasive capability was significantly higher under hypoxia. It should be underlined, once again, that the spread of melanoma typically occurs in a tumor microenvironment characterized by hypoxia. In this context, and to further characterize the potential mechanism involved, we next evaluated the metalloproteinase 2 and 9 (MMP-2, MMP-9) activity under hypoxic conditions. Indeed, the degradation of the ECM is a fundamental step for the spread of cancer cells in the tumor microenvironment [47]. To this end, we examined MMP-2 and MMP-9 activity by zimography analyses in both cell lines, wherein CAXII was transiently knocked down by siRNA. As shown in Figure 8C,D, while the activity of MMP-2 was significantly downregulated only in the SKML28 cell line, the downregulation in MMP-9 activity was also significant in the A375 cells. It should be underlined that MMP-9 is one of the most relevant MMPs involved in extracellular matrix degradation [48]. These results suggested that CAXII is particularly relevant for the spread of melanoma cells. 

## 3. Discussion

Melanoma is the deadliest form of skin cancer, and its metastasis represents the critical point for its clinical management. In this study, we aimed to further understand the molecular mechanisms that drive melanoma dissemination. By using a genetic approach based on the transient silencing (siRNA) of SMO, GLI1, and CAXII, we showed that melanoma cell migration and invasion were affected by CAXII and the Hh pathway. This was in agreement with other reports showing that Hh signaling facilitates tumor migration in several cancers [49,50,51,52]. Hypoxia is a common feature in melanoma microenvironments, and previous reports have shown that it drives melanoma progression [53]. Another characteristic of the tumor microenvironment is an acidic pH, which is associated with a higher expression of CAs, including CAXII, which is related to hypoxia and increased cell migration and invasiveness. We reported that CAXII expression was significantly higher in hypoxic A375 cells when compared with SK-MEL-28 cells. This could be explained by the fact that A375 cells are highly metastatic, while SK-MEL-28 cells are only moderately metastatic [54]. In accordance with the metastatic characteristic of A375 cells, we demonstrated that hypoxia enhanced invasion only in this cell type, supporting the previous literature showing that hypoxia promotes melanoma invasion [55]. In this paper, we also demonstrated that SMO and GLI1 siRNA diminished the expression of CAXII. This was in accordance with our previous reports that cyclopamine was able to reduce CAXII expression [45]. However, cyclopamine is a broad inhibitor of the Hh pathway, acting not only through its direct binding to the heptahelical bundle of SMO. The binding of cyclopamine is also regulated by PTCH, leading to further interactions with SMO [56]. In the present study, we genetically targeted both the receptor SMO and the transcription factor GLI1, demonstrating unequivocally the impact of both the canonical and noncanonical activation of the Hh pathway on migration/invasion and CAXII expression under hypoxia. The impact of hypoxia on SMO and GLI1 expression is controversial. Some authors have reported that hypoxia enhances the expression of the Hh ligand, SMO, and GLI1. Other reports have shown that SMO and GLI1 expression are not affected by hypoxia [57]. We here reported that SMO and GLI levels actually decreased under hypoxic conditions. However, we found out that silencing SMO and GLI1 resulted in decreased migration and invasiveness in melanoma under hypoxia. These apparently contradictory results agree to a certain extent with previous studies conducted on pancreatic cancers [58]. Indeed, despite the fact that hypoxia triggered Hh-mediated tumor stromal interactions and increased sonic secretion, the authors did not observe any effects on SMO and GLI1 expression. More interestingly, we found out that CAXII expression, cell migration, and invasiveness were inhibited by SMO siRNA and siGLI1. Once CAXII was knocked down, cell migration and invasiveness were also inhibited. Moreover, silencing CAXII resulted in the downregulation of MMP-9 and MMP-2 under hypoxia, suggesting a potential loop between hypoxia, the Hh pathway, and CAXII. The latter was upregulated by hypoxia and appeared to be positively modulated by the Hh pathway, since SMO and GLI siRNA treatment resulted in its downregulation. However, further studies are certainly needed to establish a clear connection between the Hh pathway and CAXII activity in regard to melanoma cell invasiveness.

## 4. Materials and Methods

### 4.1. Cell Cultures

SK-MEL-28 and A375 cells were kindly provided by Dr. Francesca Chiarini (University of Bologna, Bologna, Italy) and Prof. Luisa Bracci (University of Siena, Bologna, Italy), respectively. SK-MEL-28 cells were cultured in RPMI and A375 in DMEM, supplemented with antibiotics, L-glutamin, and 10 % fetal bovine serum (FBS) (Euroclone, Devon, UK). Cell lines were incubated in a humidified atmosphere (5% CO_2_, 20% O_2_, corresponding to pO_2_~140 mmHg) at 37 °C. Hypoxic experiments were conducted with the INVIVO_2_ 400 workstation (Ruskinn, Pencoed, UK), which provided a humidified and stable microenvironment through electronic control with 5% CO_2_, 2% O_2_ (corresponding to pO_2_~14 mmHg), and 37 °C temperature [59]. For transient transfection experiments, SMO and CTR siRNA sequences were purchased from Sigma-Aldrich (Saint Louis, MO, USA), whereas GLI1 and CAXII siRNA were bought from Ambion (Thermo Fisher Scientific, Cleveland, OH, USA) (Table 1). Cells were transfected with 100 nM siRNA using a NeonTM transfection system (Invitrogen, Paisley, UK). 

### 4.2. Western Blot

Western blot analyses were performed by lysing SK-MEL-28 and A375 cells with Laemmli buffer supplemented with a cocktail of proteinase inhibitors (Sigma Aldrich, Saint Louis, MO, USA). After sonication, the quantification of the total protein amounts was determined by a Micro BCA Protein Assay Reagent kit (Thermo Fisher Scientific, Cleveland, OH, USA). Proteins (30 μg protein/lane) were resolved by SDS-PAGE, transferred to 0.2 µm nitrocellulose membranes (BIO-RAD, Hercules, CA, USA), incubated for 1 h with 5% milk to block nonspecific protein sites, and then subjected to immunoblot assay. We then evaluated the expression of the proteins of interest: SMO (Santa Cruz, Dallas, TX, USA); GLI1; CAXII; and β-actin (Cell Signaling, Denver, CO, USA). Membranes were incubated with a horseradish peroxidase-conjugated appropriate secondary antibody (Cell Signaling); then, the antigen–antibody complexes were visualized using an Immunostar HRP kit (Bio-Rad Laboratories, Hercules, CA, USA). β-actin was used as housekeeping protein for protein expression. The chemiluminescence of the immunoreactive bands was detected with a CCD camera gel documentation system (ChemiDoc XRS apparatus, Bio-Rad Laboratories) and quantified with Image Lab software (Bio-Rad Laboratories). For original blots, see Appendix A. 

### 4.3. RNA Extraction and RT-qPCR

For RNA analysis, cells were cultured in normoxic or hypoxic conditions. At the appropriate time of incubation, cells were collected with euroGOLD Trifast reagent (Euroclone) and frozen at −80 °C. Total RNA was extracted following the manufacturer’s instructions. Complementary DNA was synthesized using an iScript™cDNA Synthesis Kit (Bio-Rad Laboratories), and reverse transcription quantitative polymerase chain reaction was performed using SsoAdvanced™ Universal SYBR® Green Supermix (Bio-Rad Laboratories) and an iQ™ 5 Muticolor Real-Time PCR Detection System (Bio-Rad Laboratories), as previously described [60]. Data were quantitatively analyzed using iQ™5 Optical System software (Bio-Rad Laboratories). Relative quantification was carried out using the 2^−ΔΔCT^ method [61]. Ribosomal protein L32 (L32) was used as housekeeping gene for mRNA expression. Results were expressed as fold increase in mRNA expression, with respect to the control cells.

Validated GLI1 and L32 primers were purchased from Invitrogen (Thermo Fisher Scientific, Cleveland, OH, USA), whereas SMO and CAXII were purchased from Sigma Aldrich (Saint Louis, MO, USA) (Table 2).

### 4.4. Wound-Healing Assay

SK-MEL-28 and A375 cells (either wildtype or transfected) were seeded in a culture-insert 2 well in a 35 mm µ-Dish (ibidi culture-insert 2 well, ibidi GmbH, Martinsried, Germany). After allowing the cells to attach overnight, we removed the culture insert and washed the cells with PBS to remove non-adherent cells. Cells were then washed with PBS, and fresh culture medium containing 10% FBS and AraC (2.5 µg/ml) was added. Plates were incubated in a humidified atmosphere containing 5% CO_2_ and 20 or 2% O_2_ at 37 °C for experiments performed under hypoxic conditions. Images of the wound space (10×) were taken immediately after wound initiation (timepoint 0), after 24 h (timepoint 24), and after 48 h (timepoint 48) with a microscope (Olympus IX81; Tokio, Japan). The wound parameters were quantified using ImageJ software. Data were reported as $\;\left(1-\frac{Ax}{A0}\right)\%,$ where A0 and Ax represent the empty area at timepoints 0, 24, or 48.

### 4.5. Modified Boyden Chamber

The invasion assay was performed using Boyden 48-well micro-chemotaxis chambers (Neuro Probe, Gaithersburg, MD, UK) with 8µm pore size polycarbonate polyvi-nylpyrrolidone-free nucleopore filters, precoated with 100 µl of 0.2 mg/ml Matrigel (Corning, Life Science, Corning, Tewksbury, MA, USA). Wildtype and transfected cells were plated in the upper chamber in 50 µl RPMI or DMEM with 0.1 % BSA. NIH3T3 supernatant was used as chemoattractant in the lower chamber compartment. After 24 h, cells were fixed and stained with Diff Quick (Merz-Dade, Düdingen, Switzerland) and photographed at 5× with an OLYMPUS IX81 inverted microscope. Data were expressed as number of invaded cells/field. 

### 4.6. Zymography

Culture media were collected at 24 h, centrifuged at 300× *g* for 5 min, and quantified using a Micro BCA Protein Assay Reagent kit (Thermo Fisher Scientific, Cleveland, OH, USA). Ten micrograms of each diluted media was loaded onto 1% porcine skin gelatin/10% acrylamide gels and then incubated in developing buffer with agitation at 37 °C overnight. The next day, gels were stained with Comassie R-250 (Sigma Aldrich, Saint Louis, MO, USA), pictures were taken with a ChemiDoc™ MP System (Bio-Rad, Hercules, CA, USA), and quantification was carried out using ImageJ software.

### 4.7. Statistical Analyses 

Data are presented as mean ± SEM of at least three independent experiments. Analysis of variance (ANOVA) was performed using Graph-pad Prism 7 software (San Diego, CA, USA), and statistical significance was set at *p* ≤ 0.05 (* *p* ≤ 0.05, ** *p* ≤ 0.01).

## 5. Conclusions

In conclusion, our results suggest that CAXII and the Hh pathway are implicated in melanoma invasion and may be novel and promising therapeutical targets for melanoma clinical management.

## Figures and Tables

**Figure 1 cancers-14-04776-f001:**
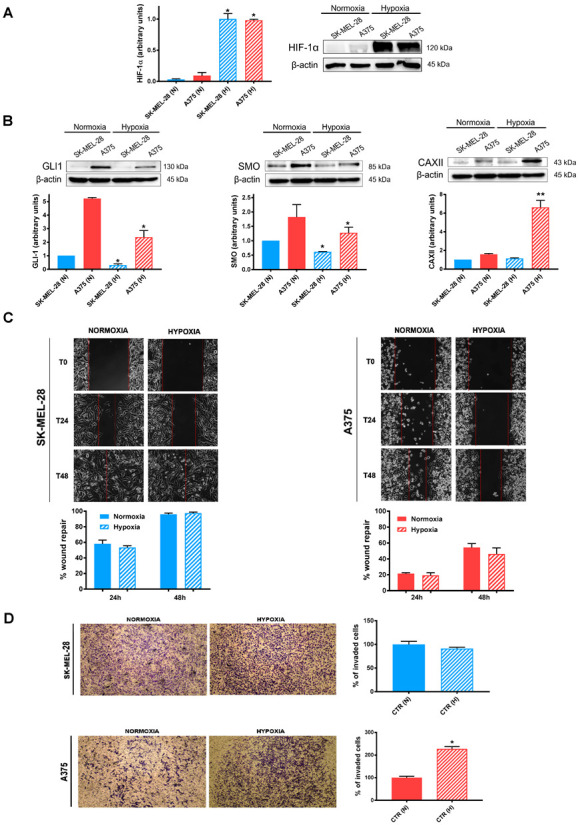
Effects of hypoxia on SK-MEL-28 and A375 cell lines. (**A**) HIF-1α; (**B**) GLI1, SMO, and CAXII western blot; (**C**) cell migration by wound-healing assay; (**D**) cell invasion by modified Boyden chamber, under normoxic (N) or hypoxic (H) conditions. β-actin was used as loading control for western blot. Blots and pictures are representative of three independent experiments. Means ± SEM are presented. (n = 3; * *p* ≤ 0.05 and ** *p* ≤ 0.01 indicate statistically significant differences).

**Figure 2 cancers-14-04776-f002:**
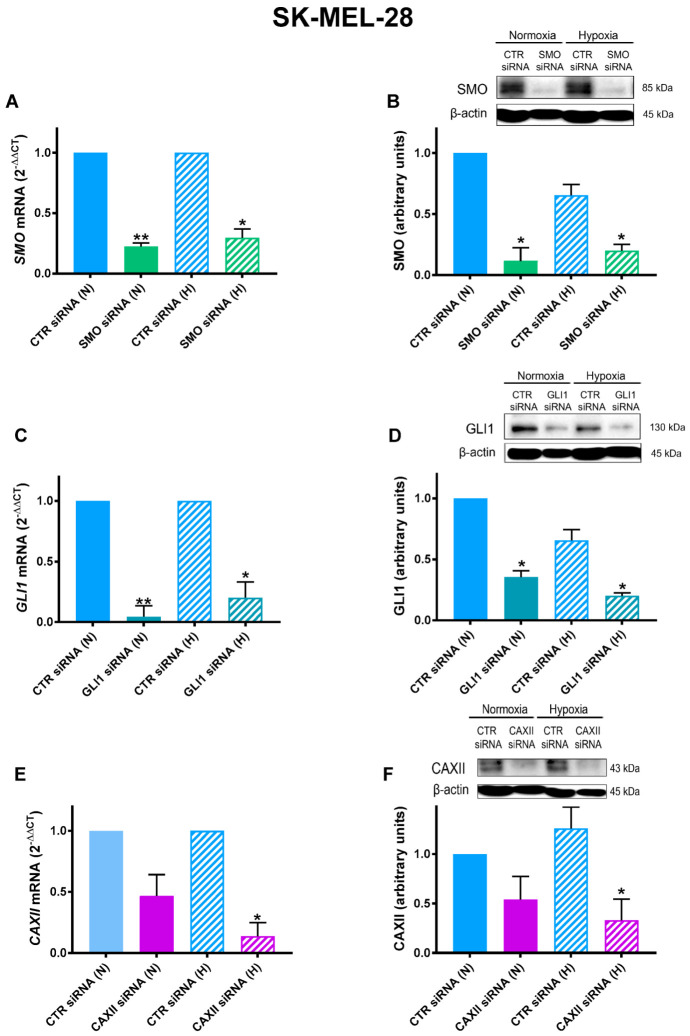
SMO, GLI1, and CAXII transient knockdown in SK-MEL-28 cell line: SMO (**A**,**B**), GLI1 (**C**,**D**), and CAXII (**E**,**F**) mRNA as determined by RT-qPCR and western blot analysis in SK-MEL-28 silenced by siRNA under normoxic (N) or hypoxic (H) conditions. β-actin was used as loading control for western blot, and L32 was used as housekeeping gene for RT-qPCR. Blots are representative of three independent experiments. Means ± SEM are presented. (n = 3; * *p* ≤ 0.05 and ** *p* ≤ 0.01 indicate statistically significant differences).

**Figure 3 cancers-14-04776-f003:**
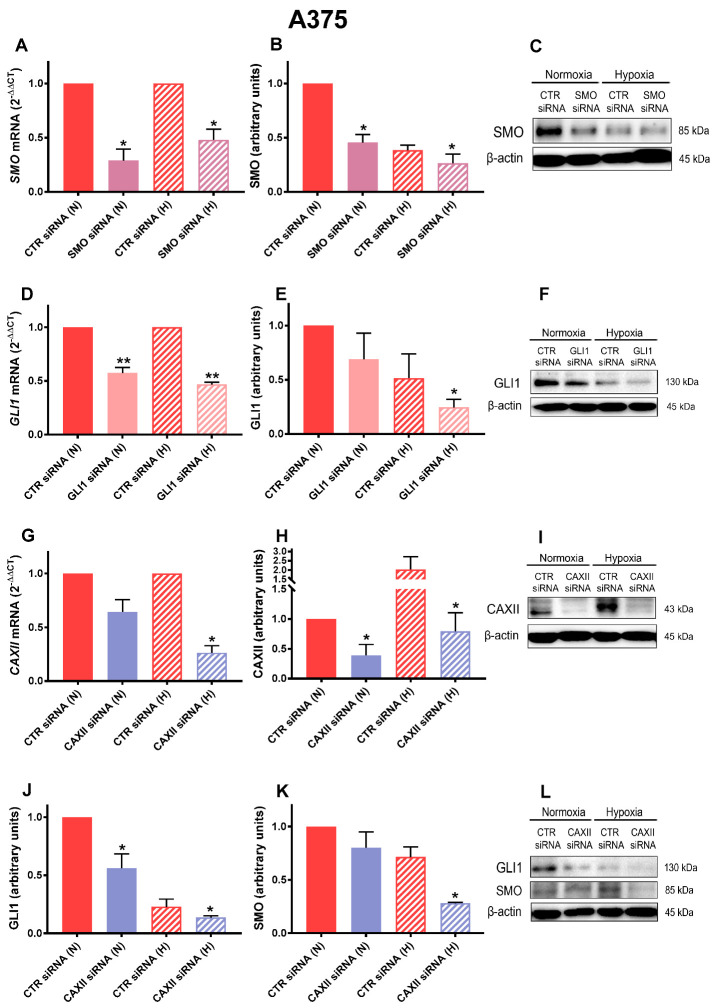
SMO, GLI1, and CAXII transient knockdown in A375. SMO (**A**–**C**), GLI1 (**D**–**F**), CAXII (**G**–**I**), and GLI1 and SMO in CAXII-siRNA-treated A375 (**J**–**L**) mRNA as determined by RT-qPCR and western blot analysis for A375 silenced by siRNA under normoxic (N) or hypoxic (H) conditions. β-actin was used as loading control for western blot, and L32 was used as housekeeping gene for RT-qPCR. Blots are representative of three independent experiments. Means ± SEM are presented. (n = 3; * *p* ≤ 0.05 and ** *p* ≤ 0.01 indicate statistically significant differences).

**Figure 4 cancers-14-04776-f004:**
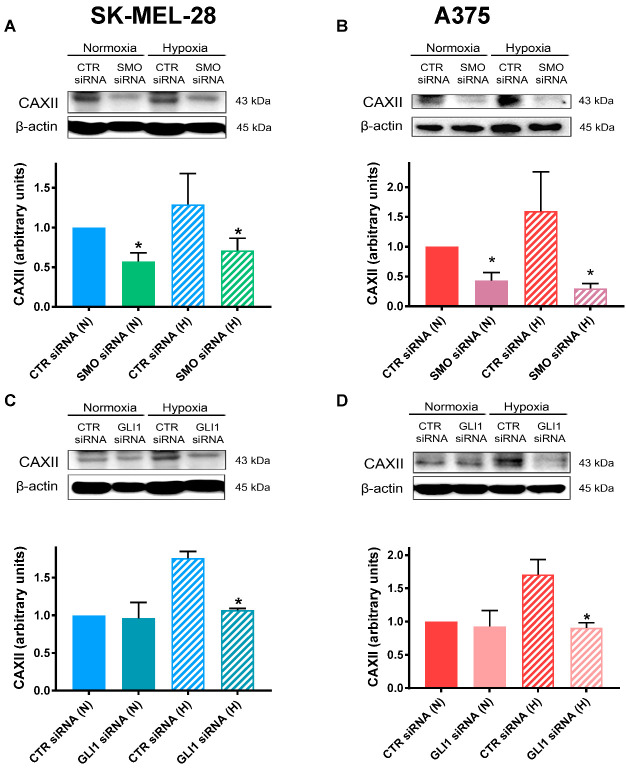
SMO and GLI1 transient knockdown reduced CAXII protein levels. CAXII protein expressions as determined by western blot analysis in SK-MEL-28 and A375 silenced with SMO siRNA (**A**,**B**) and GLI1 siRNA (**C**,**D**) under normoxic (N) or hypoxic (H) conditions. β-actin was used as loading control. Blots are representative of three independent experiments. Means ± SEM are presented. (n = 3; * *p* ≤ 0.05 indicates statistically significant difference).

**Figure 5 cancers-14-04776-f005:**
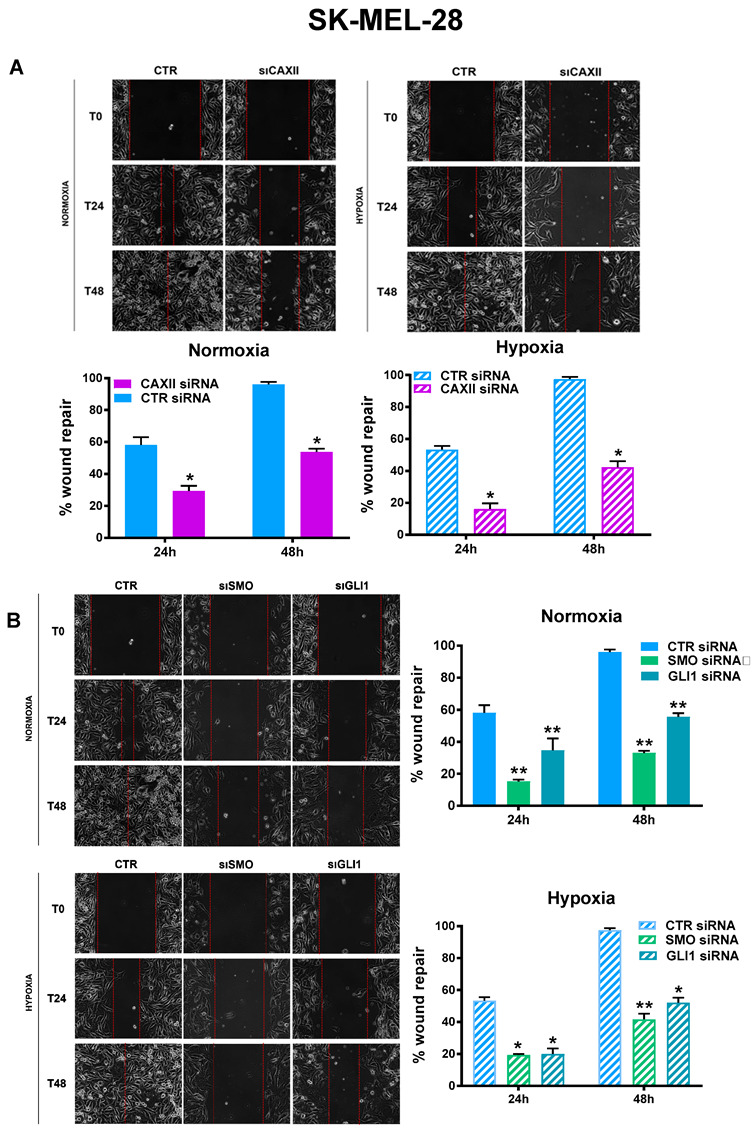
SK-MEL-28 cell migration was impaired by CAXII, SMO, and GLI1 transient knockdown. Cell migration measured by wound-healing assays under normoxic or hypoxic conditions in SK-MEL-28 cells silenced with CAXII (**A**), SMO, and GLI1 (**B**). Pictures are representative of three independent experiments. Means ± SEM are presented. (n = 3; * *p* ≤ 0.05 and ** *p* ≤ 0.01 indicate statistically significant differences).

**Figure 6 cancers-14-04776-f006:**
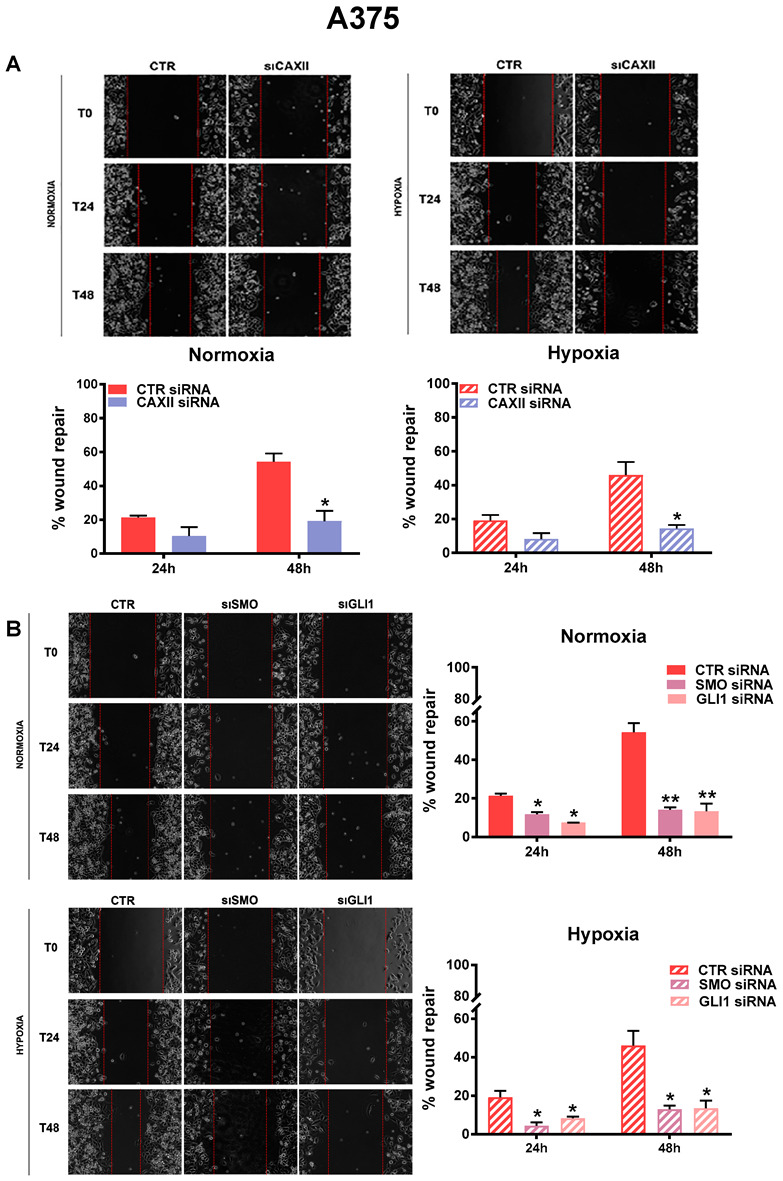
A375 cell migration was inhibited by CAXII, SMO, and GLI1 siRNA. Cell migration measured by wound-healing assays under normoxic and hypoxic conditions in A375 cells silenced with CAXII (**A**), SMO, and GLI1 (**B**). Pictures are representative of three independent experiments. Means ± SEM are presented. (n = 3; * *p* ≤ 0.05 and ** *p* ≤ 0.01 indicate statistically significant differences).

**Figure 7 cancers-14-04776-f007:**
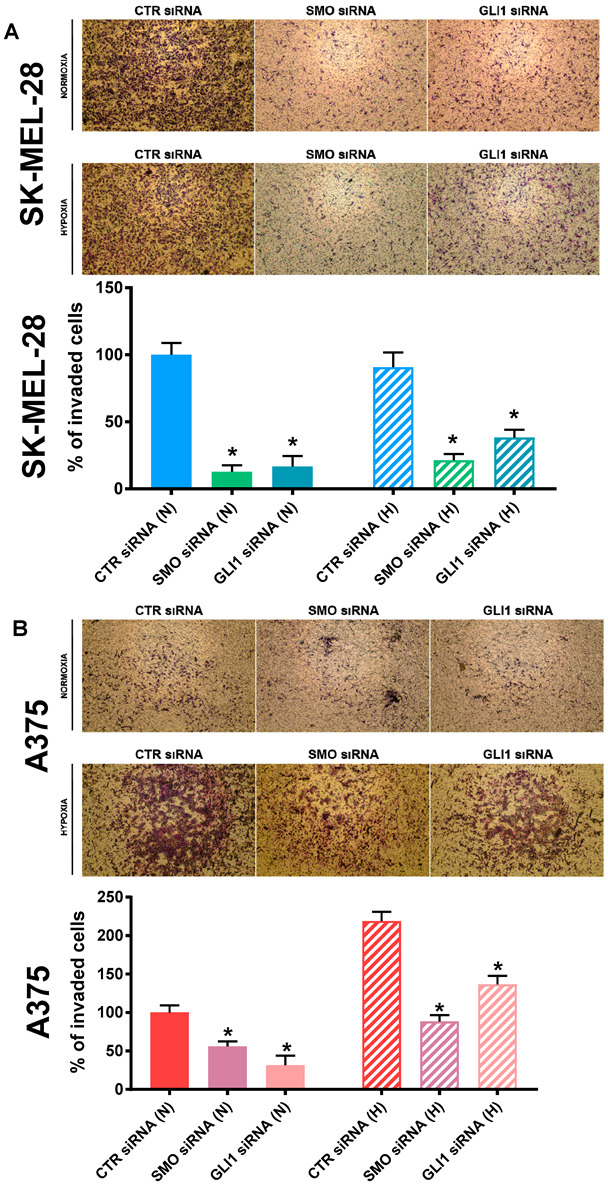
SMO and GLI1 transient knockdown impaired melanoma cell invasion. Cell invasion measured by modified Boyden chamber assay in SK-MEL-28 (**A**) and A375 (**B**) cells silenced with SMO and GLI1 siRNAs under normoxic (N) and hypoxic (H) conditions. Pictures are representative of three independent experiments. Means ± SEM are presented. (n = 3; * *p* ≤ 0.05 indicates statistically significant difference).

**Figure 8 cancers-14-04776-f008:**
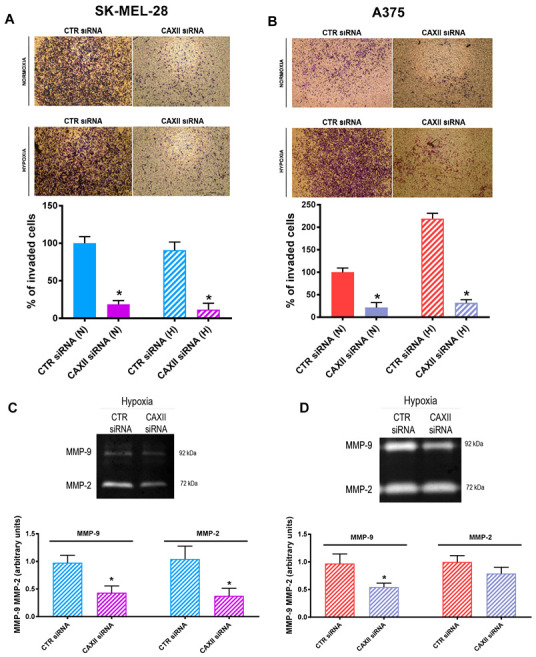
CAXII transient knockdown resulted in decreased cell invasion and MMP-2/ MMP-9 activity. Cell invasion measured by modified Boyden chamber assay in SK-MEL-28 (**A**) and A375 (**B**) cells silenced with CAXII siRNA under normoxic (N) and hypoxic (H) conditions. Zymogram assays in hypoxic SK-MEL-28 (**C**) and A375 (**D**) cells. Pictures and blots are representative of three independent experiments. Means ± SEM are presented. (n = 3; * *p* ≤ 0.05 indicates statistically significant difference).

**Table 1 cancers-14-04776-t001:** siRNA sequences.

GENE	Sense	Antisense
*GLI1*	GGAAAGCAGACUGACUGUGtt	CACAGUCAGUCUGCUUUCCcg
*SMO*	CUGUUAUUCUCUUCUACGUtt	ACGUAGAAGAGAAUAACAGcg
*CAXII*	CGGUUCCAAGUGGACUUAUtt	AUAAGUCCACUUGGAACCGcg
Scrambled	GGAUUUCUAUACGUUUAUUtt	AAUAAACGUAUAGAAAUCCcg

**Table 2 cancers-14-04776-t002:** RT-qPCR primers.

GENE	Fv	Rv
GLI1	5′ TTCCTACCAGAGTCCCAAGT	5′ CCCTATGTCAAGCCCTATTT
SMO	5′ CTTTGTCATCGTGTACTACGCC	5′ CGAGAGAGGCTGGTAGGTC
CAXII	5′ CTGCATCATGTATTTAGGGGC	5′ GAGTTGCGCCTGTCAGAAAC
L32	5′ GCTGGAAGTGCTGCTGATGTG	5′ CGATGGCTTTGCGGTTCTTGG

## Data Availability

The row data supporting the conclusions of this article will be made 502 available by the authors, without undue reservation.

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
