# Peer review of "Role of the Hedgehog Pathway and CAXII in Controlling Melanoma Cell Migration and Invasion in Hypoxia"

_cancers, 2022, doi:10.3390/cancers14194776_

Round 1

Reviewer 1 Report

In this manuscript Giuntini and colleagues explored the role of Hedgehog pathway and carbonic anhydrase XII (CAXII) in regulating melanoma cell migration and invasion under normoxic and hypoxic conditions. Authors show that inhibition of the Hedgehog pathway by silencing Smoothened or the downstream transcription factor GLI1 decrease CAXII expression in invasive melanoma cells under hypoxia.

The topic is potentially interesting, but it is too preliminary and it requires further experiments.

Major comments:

1) Authors show that SMO and GLI1 silencing downregulates CAXII expression in invasive melanoma cells under hypoxia conditions. Authors conclude that both, HH pathway and CAXII, are relevant in melanoma invasion. Similar findings about downregulation of CAXII expression upon silencing of SMO were already shown in breast cancer cell lines by the same authors in another manuscript (DOI: 10.1002/jcp.26947). To my opinion, in this study more effort should be put in trying to elucidate the connection between HH pathway and CAXII expression and activity in melanoma cell invasiveness. Another point worth addressing is whether HH pathway controls melanoma cell migration and invasion through CAXII (at least in part). In other words, the ability of HH pathway activation to promote melanoma cell migration and invasion is prevented when CAXII is inhibited?

2) In Fig. 2E-F authors show that CAXII inhibition was observed only under hypoxia, claiming that under normoxic conditions CAXII expression is almost undetectable. However, WB of SK-Mel-28 cells in Fig. 2F shows that in normoxic conditions CAXII is expressed and is not affected by siRNA. This is very confusing.

3) It would be preferable to use two independent siRNAs for each gene to minimize the possibility of off-target effects, unless authors have already chosen the best siRNAs from previous experiments.

4) Throughout the manuscript contrast of Western blots seems quite high, which may ultimately lead to perceived differences that may not be apparent in the original blot.

5) Characters in all graphs are too small, making almost impossible to read them.

6) there is an inconsistency in the text between Abstract and Introduction. In the Abstract (lines 17 -18) authors claim that “Hedgehog pathway,…, is aberrantly re-activated in melanoma and may represent a promising therapeutic target”. In the Introduction (lines 58-61) they mentioned “the role of Hh in cancer has been extensively studied [7] but little evidence exists regarding its involvement in melanoma; nevertheless, researchers agree on the fact that the aberrant Hh reactivation is responsible for melanoma cancer stem cell (CSC) proliferation, survival and self-renewal [8-11].” The sentence in the Introduction and the reported references should be revised. There are several reports supporting the involvement of HH pathway in melanoma (doi:10.1073/pnas.0700776104; doi:10.1371/journal.pone.0069064; doi: 10.3390/ph6111429; doi:10.1038/cdd.2015.56) and in melanoma cancer stem cells (doi: 10.1002/stem.1160).

7) M&M section does not report the sequences of siRNAs for SMO, GLI1, CAXII and CTR.

Minor points:

- lines 97-98 should be re-written, to read: “Therefore, in this study we used a siRNA approach to target the upstream (SMO) and the downstream (GLI1) elements of the HH pathway…..”

- Please make consistent throughout the manuscript symbols for proteins (capital), and symbols for genes and mRNA (capital and italicized).

- line 304: replace “(SSH)” with “(SHH)”.

- few typos throughout the manuscript need to be fixed.

Reviewer 2 Report

The authors tried to demonstrate the link between Hf pathway and hypoxia-inducible CAXII and their significance of melanoma phenotypes. However, as the authors described, there are many discrepancies for the regulation of above-mentioned network and to convince the readers, more experiments should be done. Major comments; 1. Throughout the paper, mechanisitic analysis is lacking, the readers would like to know the underlying mechanisms for Hh signalng pathway-mediated the regulation of CAXII expression and hypoxia-driven phenomenon.  2. Moreover, according to this manuscript, CAXII regulated the melanoma motility and invasive ability. If so, expressio of key factors (such as snail, slug, twist etc.) involving epithelial-mesenchymal transition (EMT) should be examined.  3. Fig. 1, the protein levels of HIF-1a in both SK-MEL-28 and A375 melanoma cell lines were significantly increased under hypoxia, however CAXII levels in SK-MEL-28 and A375 showed diferent pattern. Please describe about these observation.  4. Fig.1B showed that SMO and GLI1 were decreased under the condition of hypoxia, but CAXII protein levels were increased in both cell lines ( particulary in A375 cells). However, the authors showed that kockdown of either SMO or GLi1 resulted in the decrease of CAXII. If either SMO or GLI1 regulates the expression of CAXII, what mechanism causes this descrepancy?   5. Page 5, ine 137, the authors described that under normoxic conditions, CAXII expression is almost undetectable, however, Fig. 2F showed that CAXII was detected even under normoxia. This made me very confusing. Overall, there is no representativity of the expression and inducibility of CAXII in SK-Mel-28  6. Fig. 1 and 2, please describe the means of N and H.  5. Fig. 2E, the expression levels of CAXII were almost same, however, tha authors described that CAXII. 7. Fig. 2E and 3E, the mRNA levels bof CAXII etween normoxa and hypoxia were same, hwever,t the authors described that CAXII is hypoxia-inducible. 8. In Fig. 1B, the expression of CAXII was almost same irrespective under normoxic condition or hypoxia in SK-Mel-38. However, Fig. However, Fig. 4 showed that CAXIi was increased under the condition of hypoxia.  9. The authors shold examine whether the treatment with melanoma cells with cyclopamine (Hh inhibitor) can inhibit the expression of CAXII to confirm the siRNA expreriment.  10. Finally, please clarify how the hypoxia and Hf pathways affect CAXII expression and melanoma phenotype. As the authores described, clear connection among the Hh pathway, CAXII and hypoxic environment should be clarified before publish. 

Round 2

Reviewer 2 Report

The authors revised and adequately answered my comments.